# A framework for evaluating predicted sperm trajectories in crowded microscopy videos

David Hart[1]*, Kylie Cashwell[2], Anita Bhandari[1], Jayath Premasinghe[1], Cameron Schmidt[2]*

1 Department of Computer Science, East Carolina University, Greenville, North Carolina, United States of America, 2 Department of Biology, East Carolina University, Greenville, North Carolina, United States of America

☙ Shared Senior Authorship.
* hartda23@ecu.edu (DH); schmidtc18@ecu.edu (CS)

## Abstract

Since the 1980s, semi-automated sperm motility analysis of phase contrast microscopy videos has been used to measure and categorize sperm motility patterns. Motility categories are determined from various kinematic parameters such as Curvilinear Velocity (VCL) and Beat Cross Frequency (BCF). These measures ultimately rely on the quality of the tracking for each individual sperm in the microscopy video. However, common approaches to sperm tracking require sample dilution and shortening the time window of observation (less than 1 to 2 seconds) to avoid tracking errors that occur when sperm cross paths. The post-ejaculatory lifespan of sperm can exceed several hours to days in some species, and long-term adaptive changes in motility pattern may be an important distinguishing factor for predictive modeling of sperm fertilizing competence. Improving the predictive value of computer assisted semen analysis will require accurate tracking of sperm trajectories over physiologically-relevant time scales and at the high cell densities typically found in semen. In this work, we identify a framework for accurately assessing the quality of sperm trajectory tracking that is independent of standard motility measures. We utilize cell tracking metrics adapted from the more common task of tracking adherent somatic cells and propose modifications based on the unique challenges of sperm video-microscopy. We also provide a small dataset of microscopy videos that includes 340 labeled sperm trajectories to allow for future comparisons and developments. Finally, we demonstrate that variations in configuration can lead to as much as a 30% improvement on metrics, showcasing their effectiveness at analyzing tracking quality.

**Data availability statement:** The code and the dataset are publicly available at https://github.com/CAS-ReproLab/Sperm_Object_Tracking.

**Funding:** This work was supported by the Eunice Kennedy Shriver National Institute of Child Health and Human Development (R01HD110170 to CAS), as well as laboratory startup funding from the Thomas Harriot College of Arts and Sciences at East Carolina University and the East Carolina University Research and Economic Development Office. The funders had no role in study design, data collection and analysis, decision to publish, or preparation of the manuscript.

**Competing interests:** The authors have declared that no competing interests exist.

## Author summary

This report develops a computer vision framework to track individual sperm cells in crowded (high cell-density) microscopy videos, with the goal of improving automated analysis of sperm motility patterns. The task of accurately analyzing sperm movement is deceptively challenging due to the high rate of cell crossovers, an issue that has long impeded long-term tracking of sperm trajectories. Here, we introduce new evaluation metrics and provide a labeled dataset to serve as a baseline for future improvements.

## Introduction

Computer aided semen analysis (CASA) is used to assess male fertility as a function of motility pattern, cell count, and/or morphology for a wide range of species in clinical, field, and research applications [1]. Inherent difficulty arises when analyzing sperm phase contrast microscopy videos due to the tendency of cells to cross paths, resulting in misclassification of assigned trajectories by common tracking algorithms. For many years, tracking systems have relied on standardized kinematic parameters [2] to measure and classify motility properties. These kinematic parameters include curvilinear and straight-line velocity, beat cross frequency, maximum amplitude of lateral head displacement, linearity, etc. These parameters, however, are ultimately derivations of underlying tracking values and become erroneous if the tracking itself is inaccurate. In order to mediate the misclassification of trajectories, it is common for automated CASA systems to simply shorten the observation time window (video frame number) and dilute cells to well below physiological densities.

The sperm of many internally fertilizing species exhibit relatively long lifespans and undergo a series of post-ejaculatory physiological maturation steps, a process collectively known as capacitation [3]. In mammals, capacitative changes are sensitive to environmental chemistry and involve coordinated motility pattern changes, that are strongly correlated with fertility competence, such as the transition from progressive to hyperactive modes [3,4]. Despite the known importance of motility pattern transitions during capacitation, the lack of accurate tracking algorithms has not permitted direct time-series analysis of sperm behaviors, leaving a significant gap in the understanding of how changes in the motility patterns of individual sperm correspond with fertility competence during this important maturation period. Nearly all available evidence related to post-ejaculatory motility pattern changes are derived from ensemble sampling of different sperm over time with measurement windows of less than one to two seconds for trajectory analysis [5]. Importantly, recent analyses in bovine sperm that used microfluidics devices for long term monitoring suggest that phase transitions in motility pattern that oscillate over periods of several minutes may enhance search efficiency for an egg [6,7]. These observations are further supported by biophysical models that predict that phase transitions in motility pattern from high to low persistence may optimize the statistical success of random search in confined domains such as the female reproductive tract [8,9].

The next generation of sperm analysis will require long-term tracking of each sperm in high density microscopy video. Some reports have presented analysis techniques for sperm [10,11], but these reports have focused only on tracking of isolated sperm. Robust tracking techniques will need to be developed for long-term and high-density video. Given recent advances, these methods are likely to benefit from adoption of AI methods used in other video-microscopy applications such as object identification, segmentation, tracking, and pattern classification. Though these methods are common in the adherent cell tracking and lineage tracing spaces [12–15], there is currently no established baseline for determining the effectiveness of tracking results specific to the needs of sperm motility analysis. Additionally, common metrics and analyses established for general cell tracking tasks differ from the unique considerations that need to be addressed for sperm microscopy videos. Any newly proposed metrics must account for ambiguities in sperm segmentation, possible crossover of sperm from the 2D perspective, and other unique difficulties that are not common in other cell tracking problems such as signal loss in the thin pixel-sparse flagellar structure. Additionally, there are few high-quality and fully-labeled training datasets available, necessitating over-reliance on *a priori* assumptions for evaluating different tracking approaches.

In this report, we establish a framework for evaluating tracking algorithms for sperm microscopy videos. Common cell tracking metrics are adapted to work within the unique context of sperm video-microscopy. Given predicted and ground truth trajectories (represented in a simple .csv format), our bespoke Python program computes these metrics to evaluate the performance of the tracker independently of derived kinematic outputs. Establishing this framework will allow for continued development of advanced computer vision approaches that will dramatically improve this important task compared with current approaches that generally depend on simple algorithms and calculations that are sensitive to arbitrarily chosen parameter values (e.g., the smoothing window size in frame-by-frame path averaging) [2]. As a baseline, we provide five fully-labeled crowded microscopy videos of washed human sperm and perform standard trajectory analysis using the common Trackpy library [16,17]. We report the associated metrics to support iterative improvement by the computer vision community in the future. Additionally, we provide several examples using our own data and the publicly available VISEM-Tracking dataset [18] to demonstrate how poor tracking assumptions lead to erroneous kinematic outputs. We also demonstrate that improving tracking metrics improves kinematic outputs in these examples.

## Materials and methods

### Ethics statement

All procedures involving human subjects were approved by the Institutional Review Board of the Brody School of Medicine at East Carolina University (study ID: UMCIRB 20-001862). All samples were collected in accordance with the Declaration of Helsinki and with informed consent.

### Data collection

We collect a microscopy video dataset of sperm movement. The setup was designed to control the density of the sperm per video for effective trajectory labeling and analysis.

**Participants.** This study was approved by the East Carolina University Institutional Review Board and all subjects provided written informed consent prior to participating in research activities. Males ages 18-40 were included in the study. Donors with self-reported sexually transmitted infections within the previous six months were excluded. Baseline fertility was assessed via pre-screening health questionnaire and qualitative semen analysis following World Health Organization Guidelines [5].

**Chemicals and reagents.** All chemicals and reagents used in this study were obtained from Sigma Aldrich (St. Louis, MO).

**Human sperm isolation.** Semen samples underwent liquefaction at 37 °C for 30 minutes in a 5% $CO_2$ incubator. Sperm were isolated from seminal plasma by differential centrifugation in 50% isotonic percoll. Sperm were then washed by centrifugation in modified Biggers, Whitten, and Whittingham media with 3.5% bovine serum albumin. Cells were

resuspended in BWW medium for analysis. Cell counts were performed using a hemocytometer. Samples were stored in a 37 °C incubator with 5% $CO_2$ until imaging. All imaging was performed within 90 minutes of sample collection.

**Phase contrast video microscopy.**  Samples were diluted in BWW medium for imaging. Phase contrast microscopy was performed at 37 °C on a Zeiss Axervert.A1 compound light microscope fitted with a thermal stage (Carl Zeiss AG, Jena, Germany). Cell suspensions were diluted to approximately 8 million cells per mL. Diluted suspensions were added to a $20\mu m$ glass depth chamber slide (Hamilton Thorne, Beverly, MA). Phase contrast microscopy videos were obtained at 5X (0.25NA) objective or at 10X (0.25NA). Videos were recorded continuously for two minutes at nine frames per second using Zeiss Zen Lite acquisition software with accompanying time-series data collection module. Raw video files were stored as uncompressed .avi format. Videos for analysis were compressed using h.264 encoding.

## Baseline tracker and labeling

For our baseline tracking pipeline, we use the Trackpy library in Python [17]. For a baseline detector, we use the particle detector *locate* function from the Trackpy library [17]. For all experiments, we use default parameters of an 11-pixel diameter and 500 minmass. For a baseline tracker, we use the particle tracker *link* function from the Trackpy library [17]. Additionally, we use default parameters of 25 for the search range and 3 for the frame memory given the resolution, frame rate, and max speed of the sperm in our videos.

To create the fully-labeled sperm trajectory data, the sperm microscopy videos were first fed through the baseline Trackpy workflow. The resulting tracks ultimately had errors that needed to be corrected. To correct the mistakes, a tool was built using OpenCV [19] in Python that allows a human labeler to manually modify track estimates. Simple mouse and keyboard commands for adding, deleting, and merging trajectory data were included. The labeler program is included as part of our code repository provided at github.com/CAS-ReproLab/Sperm_Object_Tracking. A small dataset of five 30-second videos from different donors was hand labeled for our experiments. This provides 340 fully labeled sperm trajectories.

## Analysis of tracking algorithms - Kinematic parameter distributions

Semen characteristics are typically analyzed using a small set of standard kinematic parameters [5]. These outputs come from Computer-Aided Semen Analysis (CASA), typically performed using commercial hardware/software systems. CASA is often used to define and classify sperm motility and movement patterns [2]. CASA parameters are calculated based on the observed x and y coordinates of each sperm's nuclear centroid position in each video frame. In this work, we will focus on the following kinematic parameters.

- Curvilinear Velocity (VCL): Measures the total distance traveled along the sperm's actual curvilinear trajectory, expressed in microns per second.
- Straight-Line Velocity (VSL): Calculates the shortest distance between the starting and ending points of the trajectory, divided by the total time, also expressed in microns per second.
- Average Path Velocity (VAP): Represents the average velocity along a smoothed path of the sperm's trajectory, expressed in microns per second.

The tracking algorithms generally output distributions of these kinematic parameters, so variation in these parameters provide insights into the quality of the tracking, which the three experiments in the Results and Discussion showcase. When visualizing the distributions, we use 20 equally spaced bins between 0 and the maximum velocity value observed among all algorithms. When comparing distribution from different algorithms, we use the Earth Mover's Distance (EMD) between the predicted kinematic parameters and the ground truth distributions from the hand labeled data. This is implemented using the Wasserstein distance function in the SciPy Python library [20].

## Analysis of tracking algorithms - Direct metrics

Because of the indirect nature of the kinematic distribution comparison, this work proposes a set of metrics that allow for direct quantitative evaluation of the tracking quality.

A microscopy video is made up of individual image frames. The task of effectively tracking sperm throughout the entirety of the video can be separated into two distinct subtasks. First, the sperm must be detected in each individual frame. Second, individual sperm must be correctly identified and linked to corresponding sperm objects in subsequent frames. This process has many similarities to the general cell tracking problem in video-microscopy of adherent somatic cells. Thus, many of the metrics used to evaluate the effectiveness of cell tracking may be adapted for sperm tracking.

Of particular interest are metrics from two common benchmarks: The Multi-object Tracking Benchmark [13] and the Cell Tracking Challenge [12]. General multi-object tracking has context outside of the biological realm and well-established metrics have been used such as Multi Object Tracking Accuracy (MOTA) [21], Identification F1 Score (IDF1) [22], and Higher Order Tracking Accuracy (HOTA) [23]. These general benchmarks are calculated based on both the accuracy of the detections (true positives, false positives, and false negatives) and the number of correctly and incorrectly linked objects between frames. For cell tracking, metrics are often defined based on the spatio-temporal graph representation of the trajectories [24]. Each metric can be thought of as the number of steps needed to reconstruct the ground truth graph given the predicted graph. Metrics for the detection (DET) and linking (LNK) focus on accurate node and edge placement respectively, with these two being combined to determine the overall tracking performance (TRA). Another metric of interest is track fractions (TF) which describes the average length of trajectory that is correctly identified by a predicted trajectory. Additional details providing the mathematical definitions of each metric are provided in S1 Appendix. For this work, we implemented the tracking metrics using the py-ctcmetrics Python library [25]. For sperm tracking, the pairwise matching of prediction labels and ground truth labels is calculated based on distance between sperm centroids as discussed in the Results and discussion section.

## Results and discussion

### Tracking quality's effect on standard sperm analysis outputs

Sperm velocity distributions from CASA can reveal important insights into sperm movement patterns and functionality. For example, a bimodal distribution in a 5x magnification suggests the presence of distinct subpopulations, such as progressive, non-progressive, and hyperactivated sperm, while an exponential distribution at a 10x magnification can indicate that most sperm move slowly, with only a few reaching high velocities. Factors like sperm concentration, medium viscosity, and magnification can shift these various distributions and affect the estimated proportions of motile sperm. Analyzing and understanding these distributions helps explain how sperm movement is statistically distributed among cell populations and how selection pressures may favor certain motility traits. Kinematic parameters and their associated motility pattern classifications can be used in more complex analyses to determine if sperm are efficiently navigating their environment as a measure of fertilizing potential.

Calculating these distributions, however, ultimately depends on the reliability of the underlying tracking software. Poor tracking may give erroneous results in the calculated kinematic parameters. To demonstrate common tracking issues that may occur, we provide three experiments that illustrate poor tracking assumptions and the effects on the calculated kinematic parameters.

**Experiment 1: A poorly configured tracker.** Many tracking algorithms have default configurations that are used that control properties such as maximum linking distance between frames. The default parameters that are used are unlikely to be effective across the many types of sperm microscopy videos that are captured. However, researchers and technicians may not have access to meaningful ways to tune these parameters based on their specific data depending on the acquisition and analysis software configuration.

For this first experiment, we demonstrate the effect of a poorly tuned tracker on one of our microscopy videos. For illustrative purposes, we intentionally handicap our baseline approach using the same Trackpy detect and tracking methods, but using a search range of 7 pixels (the default parameter) instead of 25 which we have previously determined to be optimal. This greatly limits the algorithm's ability to track fast-moving sperm. Many inaccuracies in the tracking may occur from just one poorly optimized hyperparameter. In particular, because of the frequently dropped fast-moving sperm, the velocity distributions become highly skewed, drastically misrepresenting the true underlying behavior of the sperm population. An example of the difference between the poorly predicted velocity distribution and the actual velocity distribution is shown in Fig 1. This example is taken from video 3 in our dataset.

To quantify this difference, we use Earth Mover's Distance (EMD) between the two distributions. For this example, the EMD in VAP, VSL, and VCL between the poor prediction and the actual labeled data is 8.20, 5.43, and 7.97 respectively. In comparison, the properly configured baseline has EMD values of 1.73, 1.48, and 1.89 respectively. This experiment demonstrates the need for more effective tools and metrics for evaluating the quality of the underlying tracking and improving confidence in predicted kinematic outputs.

**Experiment 2: Short vs. Long timescale trajectories.**  It is a common practice to collect or aggregate sperm microscopy tracking data over very short time intervals (~1-3 seconds) [2]. This practice is necessary to eliminate tracking errors that occur due to sperm moving near or over each other in the 2D image projection. Though this practice may

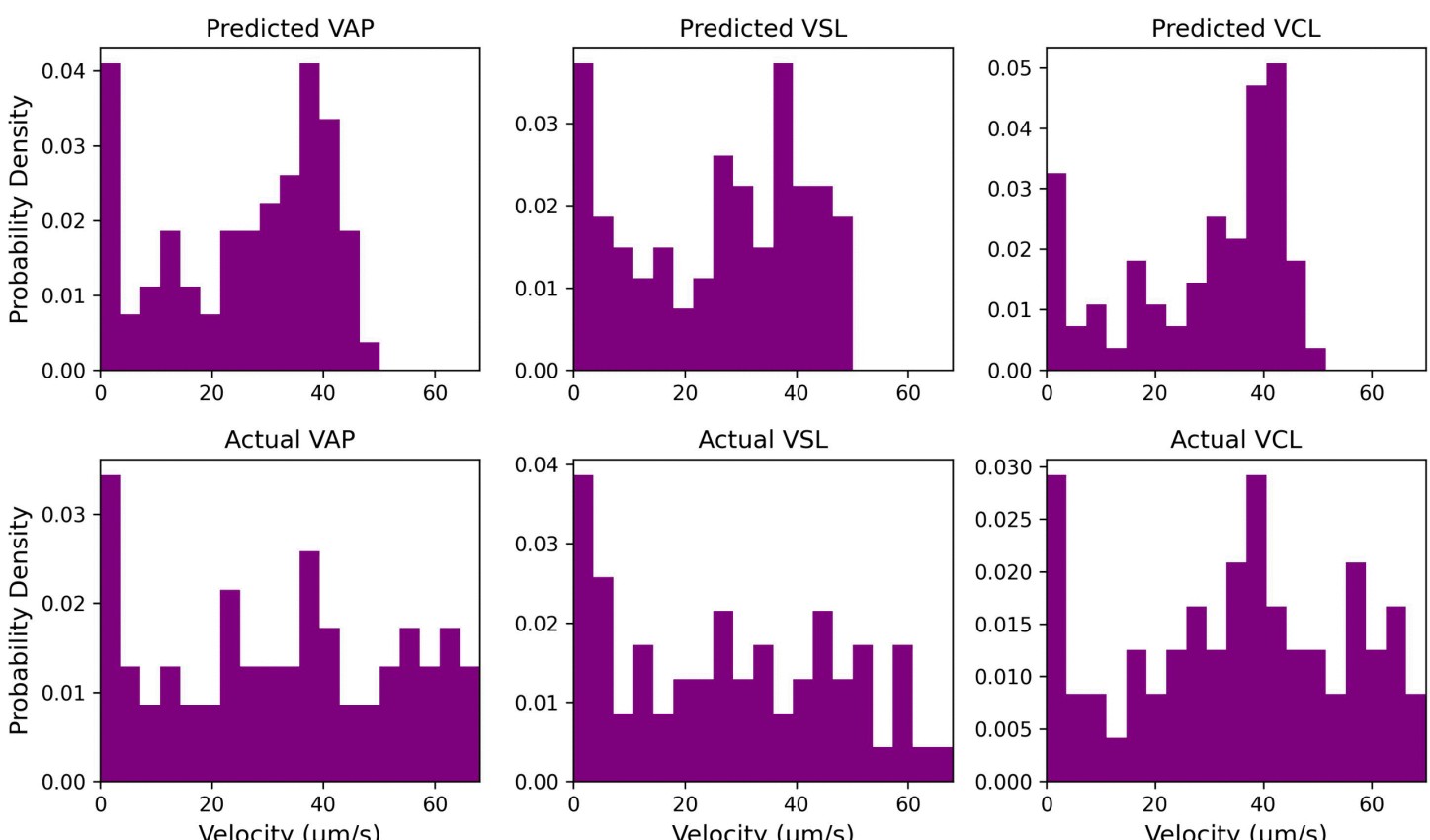

**Fig 1**. **Velocity distribution comparison.** Example distributions for the Average Path Velocity (VAP), Straight Line Velocity (VSL), and Cuvilinear Velocity (VCL). If a particular tracking algorithm is unsuited to capture high-velocity or crowded trajectories (top), it will misrepresent the motility properties of the sperm population (bottom).

alleviate some tracking errors, it comes at the cost of being unable to capture more complex, long timescale behaviors of sperm [6,7]. Here, we analyze our labeled dataset which includes longer videos in comparison with standard CASA analysis (i.e., 30 seconds), enabling comparative analysis of both the short and long term movement behaviors among the same group of sperm.

Two second subsets of the thirty second videos were compared and kinematic parameters were calculated. The 30-second video provides much richer information for understanding more complex sperm movement behaviors. For example, on the short timescale, progressively moving sperm look very similar, however, on longer timescales oscillating patterns are observed with various phases and symmetries. As expected, the 2-second subsets fail to capture the underlying long-term behavior of the sperm. A visual comparison between short-term and long-term trajectories is provided in Fig 2.

Additionally, we compared the output parameter distributions between the short-term and long-term trajectory ensembles in Fig 3. As indicated, shorter videos may not capture long-term behavior of sperm that would more accurately reflect their motility properties. This effect highlights limitations in the restricted timescales required by tracking issues in standard CASA, as well as challenges associated with the kinematic parameter calculations themselves. Because VAP, VSL, and VCL are generally represented using time-averages, the values tend to converge despite the visually apparent differences in motility behaviors. Though some useful additional information may be gleaned from analysis of kinematic parameter variance, this information is not part of the standard CASA analysis framework and is rarely reported [5].

**Experiment 3: Bounding Boxes vs. Centroids.** Multi-object tracking has seen a large amount of interest in the computer vision field [26,27] and many AI based approaches are revolutionizing videomicroscopy analysis [15,28,29]. AI based methods are in the early stages of application in solving the sperm tracking problem [11,30]. Several rely on the publicly available VISEM-Tracking dataset [18] which used manual annotation to define bounding boxes around sperm nuclei, similar to the YOLO frameworks [31] used frequently in other areas of computer vision.

As more of these bounding box trackers are used, an important issue arises. Since the bounding boxes are only approximately centered on the sperm nuclear centroid position, the path of the bounding boxes may not accurately portray the movement of the sperm, resulting in misleading positional data for sperm kinematic parameter calculations. A visualization of the bounding box movement and the sperm nuclear centroid movement is shown in Fig 4.

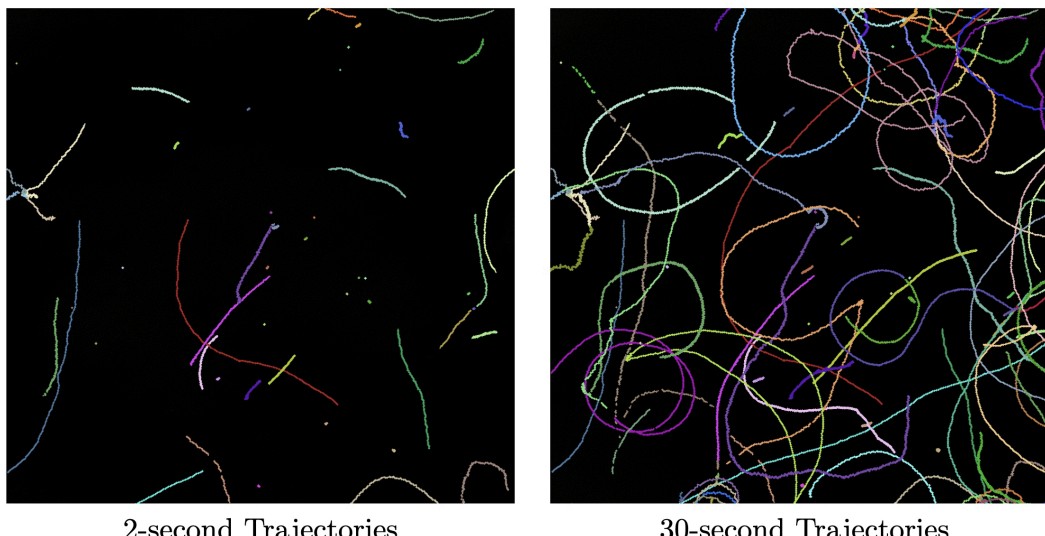

2-second Trajectories 30-second Trajectories

**Fig 2**. **Sperm tracking example.** Example visualization of sperm paths for a single video. In the short term, sperm trajectories present similar information (left). In contrast, long-term tracking highlights evolving differences in motility behaviors (right).

**Fig 3**. **2-second vs 30-second trajectory distributions.** Short-term trajectory analysis tends to underestimate convergent parameter distributions that result from longer time-averaging.

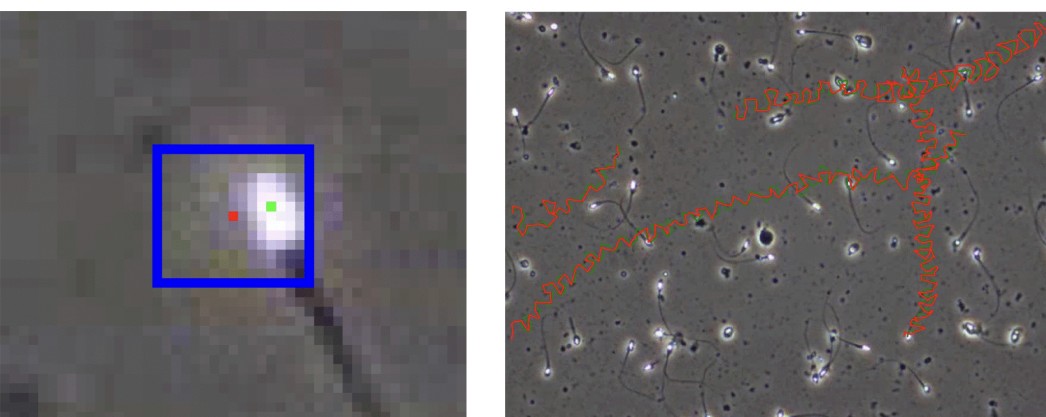

**Fig 4**. **VISEM-tracking corrections.** The publicly available and commonly used VISEM-Tracking dataset uses manually applied bounding boxes to indicate the location of sperm nuclei. However, taking the center of these bounding boxes (shown in red) does not necessarily correspond to the centroid of the sperm heads (shown in green), calculated using the Gaussian Blob method from the Python Trackpy library [17]. Trackpy was used to correct the nuclear centroid locations given by VISEM-Tracking, making the measured motility behavior much more accurate.

For this experiment, we used the VISEM-Tracking dataset and extracted head centroid placement in each bounding box. This was done using the baseline Gaussian Blob method provided by Trackpy followed by taking the presented $x,y$ coordinates that fell within each bounding box. If zero or multiple coordinates were present in a box, the bounding box center was taken to avoid ambiguity, allowing for correction of nuclear centroid placement throughout the entire modified VISEM-Tracking dataset.

As expected, the bounding box center vs the head centroids give different velocity distributions. An example is given in Fig 5. The average EMD for each kinematic parameter across the 20 video dataset was also computed. The VSL EMD remained small with an average value of 1.1. This was expected because VSL is simply the time-averaged Euclidean distance from the starting location and ending location in the video segment. In this case, each sperm still approximately started and ended at the same positions. The VAP and VCL, however, saw dramatic changes between the corrected and non-corrected versions, with average EMD values of 11.8 and 53.0 respectively. These kinematic parameters were expected to be more susceptible to tracking method because even small inaccuracies in position accumulate as noise over time.

## Evaluating sperm detection and tracking

As we have demonstrated with these simple examples, there is a substantive need to establish standardized metrics that can quantitatively compare and evaluate the performance of sperm tracking algorithms. We now turn to discussion of some candidate metrics and justify their use for the unique challenges associated with sperm video-microscopy.

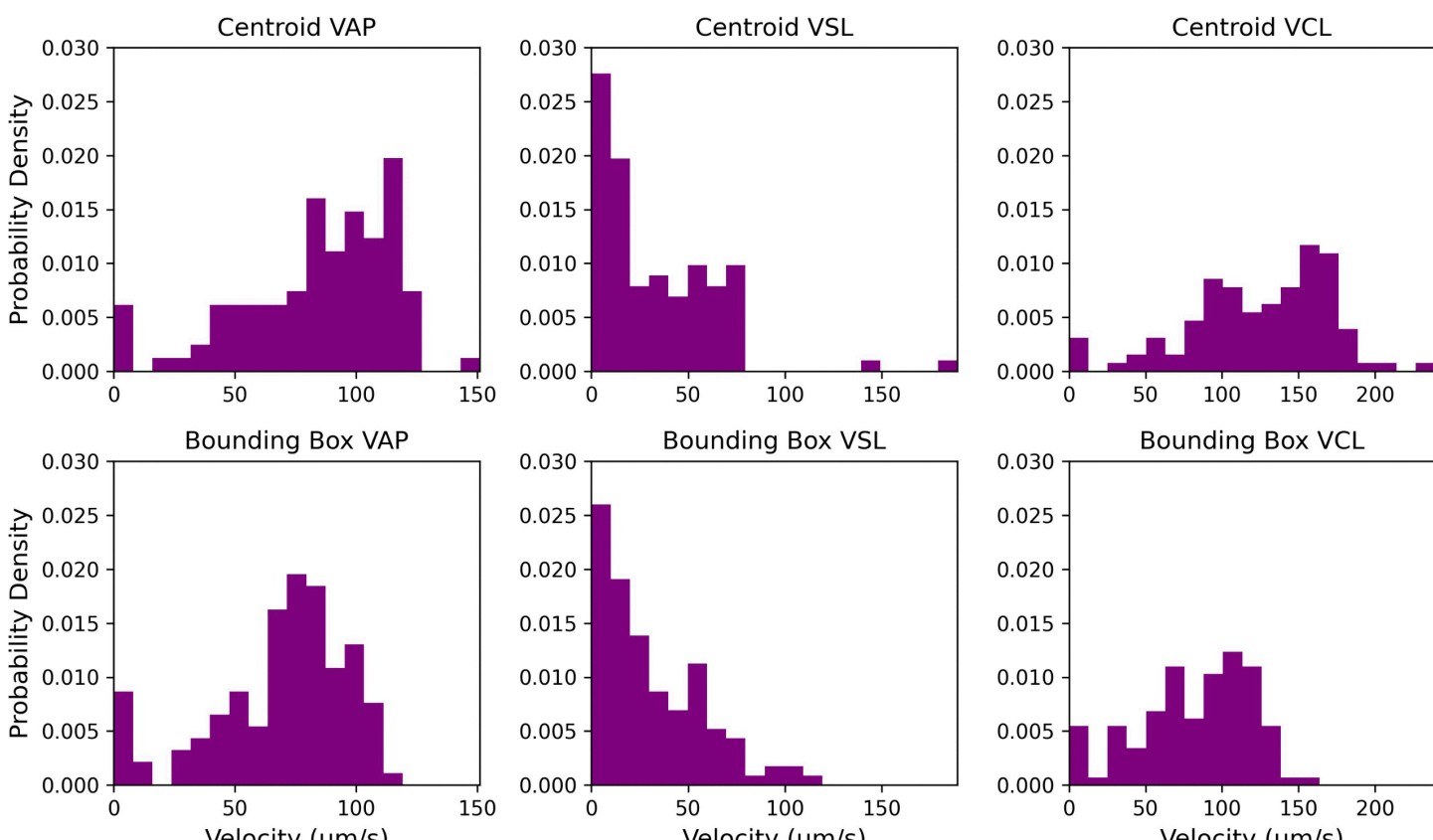

**Fig 5**. **Velocity distribution comparison between bounding box centers vs Trackpy centeroids.** Using Trackpy to correct the centers of the bounding boxes leads to differences in the resulting kinematic parameter distributions.

In the Methods section, we described common cell and multi-object tracking metrics. These common tracking metrics provide a foundation for evaluating and comparing the quality of sperm tracking algorithms. Notably, sperm tracking does present some unique differences compared with standard cell tracking that must be considered and properly accounted for. We will describe specific considerations for each of these differences in the following subsections.

**Unique considerations for evaluating sperm detection.**  Unlike most cells, sperm are incredibly thin, typically ranging from 0.8 to 0.01 $\mu m$ along the flagellar length. Even at high magnifications and resolutions, the flagellar width may only occupy a few pixels at its width, but may be many pixels long. In most somatic cell-tracking detection metrics, it is assumed that a segmentation or a bounding box is provided during the detection process. This is needed to accurately compute the Jaccard similarity scores (Intersection over Union) and match predicted trajectory IDs to the reference IDs [12]. However, relying on these common similarity scores for sperm trajectories will likely lead to inconsistent association scores due to the ambiguity in labeling of the sperm pixels. Additionally, there are multiple confounding factors that can mislead the detection process when compared to somatic cell-tracking algorithms:

- Variations in brightness of pixels within a video can make some sperm more visible than others and make sperm outlines unclear.
- Dust, dead cells, and other debris in the video may appear like sperm, but ultimately do not move within the video.
- Sperm may enter and exit the frame, and sperm near boundaries of the video may be partially missing in shape.
- Sperm pixels can overlap as they swim above or below each other, making them appear as one or more sperm of irregular shape.
- Labeling programs may be inconsistent in the use of points, bounding boxes, or segmentations for identifying sperm, as well as differences in the labelers that are used to isolate the whole cells, flagellum, or nucleus only.

Examples of the unique computer vision challenges associated with phase contrast microscopy videos of sperm are shown in Fig 6.

To combat these issues and establish correct predicted-to-actual ID linking for all types of sperm trajectory videos, it is reasonable to represent sperm detection using a single point rather than a segmentation or bounding box. The x,y coordinate is approximately centered on the head of the sperm which we call the sperm *centroid* (since most of the mass is located in the sperm nucleus). Localization between the actual and predicted point can be quickly computed using

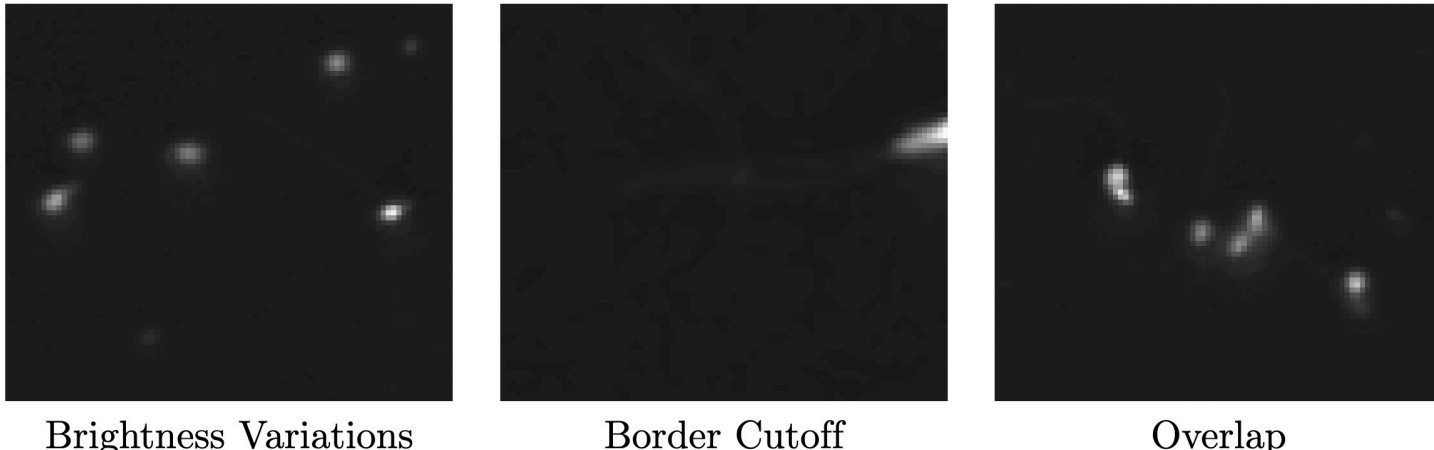

|  Brightness Variations  |  Border Cutoff  |  Overlap  |

**Fig 6**. **Sperm detection issues.** Example issues with detection of sperm. Variations in brightness and dead cells in view (left), partially missing sperm near boundaries (middle), overlap and irregular shapes (right).

the Hungarian algorithm with a distance cutoff. This association step based purely on centroid location is vital for effective evaluation since all standard tracking metrics require a consistent pairing mechanism between prediction labels and ground truth labels. For this work, the standard multi-object metrics (MOTA, IDF1, HOTA) and cell tracking metrics (DET, LNK, TRA, TF) are modified to include this distance-based association step. Additional details are described in the Methods section and S1 Appendix.

**Unique consideration for evaluating sperm tracking.** Once sperm have been identified in all frames of the video, they must be properly linked together to make a complete tracking of individual sperm. Some approaches may also rely on detections in previous frames to update detections in future frames and link in a forward fashion. Regardless of the approach, the goal is to assign each sperm with a unique identifier that it keeps throughout the whole microscopy video. Doing so enables drawing the trajectory path of each sperm and calculating the kinematic parameters.

Tracking sperm presents unique challenges compared to most cell tracking problems. First, many sperm move very quickly compared to the spatial resolution of most microscopy videos. Even during high FPS recordings, some sperm may move multiple pixels per frame. For tracking algorithms, this can cause frequent dropout and relabeling of sperm in subsequent frames. Second, as mentioned previously, sperm paths can cross over each other as one sperm swims above another. This crossover can cause an incorrect path labeling, where the two sperm swap trajectory labels in subsequent frames. These two issues are illustrated in Fig 7.

In addition to these challenges, immotile sperm and debris may be of interest for automated counting, but are not of interest for more computationally intensive motility analysis and may mislead performance benchmarking of the tracking algorithm. Immotile sperm are easier to identify in each subsequent frame compared with motile sperm and may disproportionally inflate tracking metrics for a given algorithm. Additionally, some nonmoving dust and debris may look like sperm, and be incorrectly labeled in reference datasets. Finally, failure to capture long-timescale movement patterns is another major challenge for sperm motility analysis. Improved analysis of long-timescale behavior is essential to better understand the complex process of capacitation, because it involves heterogeneous context-dependent changes in motility pattern that are either not captured, or are misled, by current motility analysis algorithms [3,4,32,33].

Effective tracking algorithms must be able to account for these confounding factors and capture objects moving at variable speeds over longer timescales. When labeled ground truth data is present, the standard tracking metrics are used in

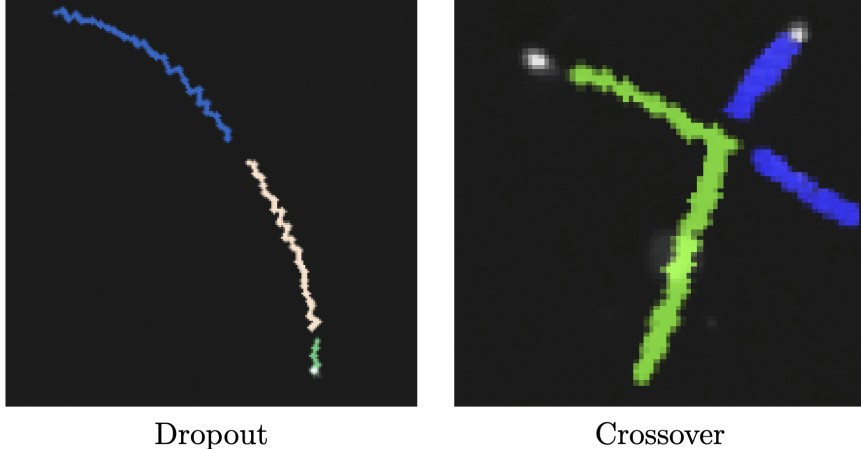

Dropout　　　　　　Crossover

**Fig 7**. **Sperm tracking issues.** Example visualization of sperm paths issues. A single sperm may appear to dropout for a few frames, causing the algorithm to think that new sperm are appearing in and out of view (left). Two sperm that intersect may cause the path of one sperm to be assigned to another, causing a straightline path to be incorrectly marked as jagged (right).

this report. However, we also propose an additional filtered version of these metrics which removes non-moving objects and very-short trajectories. For both the ground truth and predicted paths, a pre-processing step was used to remove all trajectories that do not spatially extend past a small radius of ($\epsilon$). This removes all trajectories for nonmoving or nearly nonmoving sperm, giving more weight on the metric to faster moving sperm and longer-term trajectories.

**Evaluation on the dataset.**

With the evaluation process established in the previous section, the detection and tracking metrics are presented for a baseline tracking method with a small dataset of five microscopy videos that were collected from separate healthy human sperm donors. For comparison, we also present results from a re-analysis of the publicly available VISEM-Tracking dataset [18]. In the following subsections, we present the outputted metrics for the baseline Trackpy method. Additionally, we demonstrate how the metrics can indicate the reliability of standard motility outputs.

**Metric computations.** For our dataset containing videos with manually corrected trajectories, we use the baseline Trackpy library and report the common cell tracking metrics for five videos. The ID association process described previously was used. We compared results with and without filtering the non-moving objects in the calculations. The results are shown in Table 1. The results for VISEM-Tracking are given in Tables 2 and 3. For all metrics, a higher score indicates better performance and 1.0 is the highest score. The outputted metrics serve as a baseline for future tracking algorithms to compare against.

**Relating tracking to motility outputs.** The presented metrics can determine the effectiveness of the tracking, where improving on the metrics also improves confidence in the calculated motility parameters. To demonstrate this, we conduct the following analysis.

First, we present the detection and tracking metrics, comparing the suboptimal (default) configuration to the optimal configuration for the Trackpy approach after filtering. These results are presented in Table 4. The data support the conclusion that the default configuration significantly under-performs in every metric, indicating that the metrics adequately represent the ability to track fast-moving objects.

Second, we present the computed motility outputs for the predictions from the suboptimal and optimal configurations, as well as the actual outputs from the hand labeled data. These distributions are shown in Fig 8. As can be seen, these motility outputs are drastically affected by the tracking parameters. Specifically, the suboptimal approach cannot track the trajectories of fast moving sperm, greatly skewing the resulting distributions, ultimately affecting the predicted motility categories.

We also output Earth Mover's Distance (EMD) scores for each kinematic parameter for the suboptimal and optimal approach when compared to the ground truth. These scores are provide in Table 5. As expected, the handicapped approach has larger EMD scores, indicating its inability to provide accurate kinematic parameters. Overall, this indicates the wide variation in possible tracking consistency between algorithms and configurations and the need for open tracking metrics to evaluate the reliability of these motility values.

**Table 1**. **Detection and tracking results.**

| Type\Metrics | Count | DET | LNK | TRA | TF | MOTA | IDF1 | HOTA |
|---|---|---|---|---|---|---|---|---|
| Unfiltered | 340 | 0.980 | 0.748 | 0.950 | 0.887 | 0.886 | 0.857 | 0.876 |
| Filtered | 264 | 0.959 | 0.749 | 0.932 | 0.872 | 0.814 | 0.811 | 0.834 |

Table notes: Common multi-object and cell-tracking metrics computed for the sperm trajectories in the dataset. For all metrics, a higher score indicates better performance and 1.0 is the highest score. The baseline Trackpy approach is measured against the human-labeled and corrected data. Additionally, the results are also compared before and after filtering the predictions and labels to exclude nonmoving objects, better capturing the trackers ability to follow fast moving sperm.

**Table 2**. Detection and tracking results - VISEM-Tracking (Bbox).

| Video | Count | DET | LNK | TRA | TF | MOTA | IDF1 | HOTA |
|---|---|---|---|---|---|---|---|---|
| 11 | 57 | 0.766 | 0.881 | 0.781 | 0.605 | −0.366 | 0.472 | 0.528 |
| 12 | 117 | 0.773 | 0.748 | 0.769 | 0.392 | 0.583 | 0.526 | 0.555 |
| 13 | 73 | 0.854 | 0.855 | 0.854 | 0.570 | 0.779 | 0.770 | 0.760 |
| 14 | 8 | 0.642 | 0.974 | 0.685 | 0.748 | −2.374 | 0.331 | 0.425 |
| 15 | 42 | 0.668 | 0.882 | 0.696 | 0.827 | −1.269 | 0.422 | 0.499 |
| 19 | 51 | 0.358 | 0.846 | 0.421 | 0.449 | −4.240 | 0.176 | 0.282 |
| 21 | 80 | 0.808 | 0.918 | 0.822 | 0.653 | −0.251 | 0.534 | 0.574 |
| 22 | 37 | 0.681 | 0.958 | 0.717 | 0.764 | −1.835 | 0.334 | 0.434 |
| 23 | 11 | 0.133 | 0.867 | 0.228 | 0.880 | −6.469 | 0.210 | 0.342 |
| 24 | 111 | 0.821 | 0.844 | 0.824 | 0.511 | 0.481 | 0.599 | 0.633 |
| 29 | 11 | 0.507 | 0.779 | 0.543 | 0.545 | −2.087 | 0.310 | 0.411 |
| 30 | 54 | 0.801 | 0.918 | 0.816 | 0.716 | −0.289 | 0.514 | 0.571 |
| 35 | 63 | 0.669 | 0.798 | 0.686 | 0.330 | −0.958 | 0.215 | 0.301 |
| 36 | 135 | 0.810 | 0.897 | 0.821 | 0.711 | −0.057 | 0.440 | 0.511 |
| 38 | 46 | 0.906 | 0.937 | 0.910 | 0.693 | 0.569 | 0.710 | 0.744 |
| 47 | 24 | 0.534 | 0.841 | 0.574 | 0.458 | −2.505 | 0.295 | 0.374 |
| 52 | 48 | 0.693 | 0.874 | 0.716 | 0.739 | −0.992 | 0.402 | 0.476 |
| 54 | 79 | 0.787 | 0.921 | 0.804 | 0.676 | −0.569 | 0.486 | 0.550 |
| 60 | 52 | 0.780 | 0.849 | 0.789 | 0.761 | 0.089 | 0.617 | 0.640 |
| 82 | 72 | 0.832 | 0.880 | 0.838 | 0.782 | 0.313 | 0.653 | 0.685 |

Table notes: Resulting metrics of our baseline approach when run on the VISEM-Tracking dataset using the original bounding box centers as reference.

**Table 3**. Detection and tracking results - VISEM-tracking (centroid).

| Video | Count | DET | LNK | TRA | TF | MOTA | IDF1 | HOTA |
|---|---|---|---|---|---|---|---|---|
| 11 | 57 | 0.852 | 0.907 | 0.859 | 0.718 | −0.275 | 0.496 | 0.558 |
| 12 | 117 | 0.950 | 0.866 | 0.939 | 0.722 | 0.819 | 0.599 | 0.669 |
| 13 | 73 | 0.965 | 0.900 | 0.957 | 0.818 | 0.931 | 0.835 | 0.868 |
| 14 | 8 | 0.645 | 0.977 | 0.688 | 0.855 | −2.373 | 0.343 | 0.427 |
| 15 | 42 | 0.758 | 0.976 | 0.786 | 0.851 | −1.311 | 0.432 | 0.507 |
| 19 | 51 | 0.477 | 0.884 | 0.530 | 0.555 | −4.095 | 0.195 | 0.306 |
| 21 | 80 | 0.855 | 0.934 | 0.865 | 0.780 | −0.229 | 0.532 | 0.588 |
| 22 | 37 | 0.705 | 0.974 | 0.740 | 0.824 | −1.883 | 0.335 | 0.438 |
| 23 | 11 | 0.152 | 0.992 | 0.261 | 0.842 | −7.420 | 0.211 | 0.343 |
| 24 | 111 | 0.901 | 0.728 | 0.878 | 0.678 | 0.589 | 0.645 | 0.689 |
| 29 | 11 | 0.727 | 0.993 | 0.762 | 0.918 | −1.683 | 0.401 | 0.498 |
| 30 | 54 | 0.857 | 0.965 | 0.871 | 0.884 | −0.285 | 0.526 | 0.597 |
| 35 | 63 | 0.799 | 0.855 | 0.806 | 0.530 | −0.727 | 0.265 | 0.368 |
| 36 | 135 | 0.861 | 0.806 | 0.854 | 0.729 | −0.012 | 0.441 | 0.521 |
| 38 | 46 | 0.950 | 0.968 | 0.953 | 0.855 | 0.624 | 0.720 | 0.767 |
| 47 | 24 | 0.636 | 0.956 | 0.678 | 0.825 | −2.417 | 0.317 | 0.413 |
| 52 | 48 | 0.760 | 0.873 | 0.774 | 0.793 | −0.917 | 0.419 | 0.493 |
| 54 | 79 | 0.843 | 0.962 | 0.858 | 0.805 | −0.502 | 0.499 | 0.569 |
| 60 | 52 | 0.907 | 0.977 | 0.916 | 0.926 | 0.231 | 0.667 | 0.695 |
| 82 | 72 | 0.919 | 0.923 | 0.920 | 0.867 | 0.397 | 0.684 | 0.722 |

Table notes: Resulting metrics of our baseline approach when run on the VISEM-Tracking dataset using the corrected centroids as reference.

## Implications for future sperm analysis

Recent advances in computer vision are revolutionizing fields that rely on automated image analysis such as healthcare, self-driving vehicles, and manufacturing. Adaptation of computer vision algorithms and machine learning models from

**Table 4**. **Detection and tracking ablation.**

| Config\Metrics | DET | LNK | TRA | TF | MOTA | IDF1 | HOTA |
|---|---|---|---|---|---|---|---|
| Suboptimal | 0.692 | 0.548 | 0.673 | 0.610 | 0.608 | 0.647 | 0.691 |
| Optimal | 0.959 | 0.749 | 0.932 | 0.872 | 0.814 | 0.811 | 0.834 |

Table notes: Comparison of tracking and detection metrics between the suboptimal configuration and the optimal configuration on the filtered data of 264 trajectories. The suboptimal configuration is the default configuration for Trackpy parameters which fails in tracking fast-moving objects, showcasing the metrics effectiveness in the sperm tracking task with differences ranging from 15% to 30% between configurations.

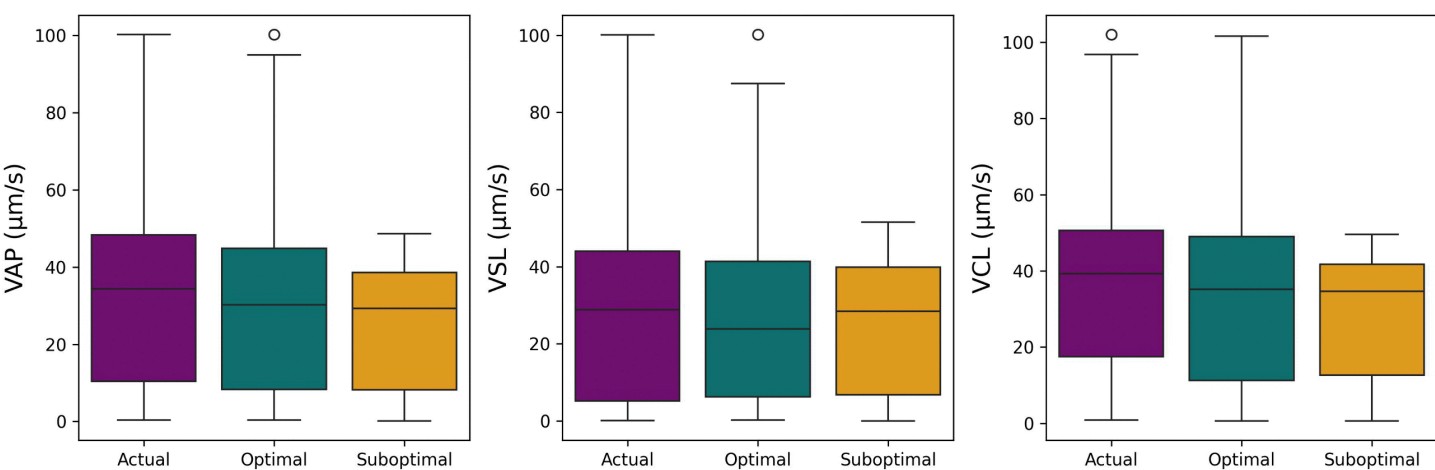

**Fig 8**. **Motility outputs.** The outputs for VAP, VSL, and VCL on the suboptimal configuration, optimal configuration, and actual hand labeled and filtered data for across the set of sperm trajectories. As can be clearly seen, the suboptimal approach fails to capture any of the higher velocities in the dataset.

**Table 5**. **Earth mover's distance.**

| Parameter | Suboptimal | Optimal |
|---|---|---|
| VAP | 7.21 | 3.40 |
| VSL | 3.38 | 2.79 |
| VCL | 7.19 | 3.59 |

Table notes: Comparison of Earth Mover's Distance between the suboptimal and optimal approach when compared to the actual hand labeled data.

these fields has dramatically benefited microscopy applications in cell biology due to improved cell location, segmentation, and tracking but has been largely restricted to adherent somatic cells in subculture. Though CASA has proven extremely useful in basic research and agricultural breeding programs, its use in clinical andrology laboratories remains limited because it does not yet outperform manual semen analysis by trained lab technicians. Here, we identified several key issues with current sperm tracking algorithms and kinematic calculations employed in CASA. One major barrier to improvement is a lack of standardized metrics to compare tracking performance that conform to the unique challenges inherent in phase contrast microscopy of motile sperm. We suggest several methods, borrowed from other applications in computer vision, to facilitate quantitative comparison of sperm tracking program performance. Additionally, we highlight important issues that arise from inaccurate labeling methods in common training datasets, such as the bounding box method used in the publicly available VISEM-tracking dataset which unintentionally resulted in significant short timescale noise [18]. Our report highlights two major issues: 1) CASA could be much more powerful than it currently is but requires

specialized metrics to compare performance among algorithms and models, and 2) there is a conspicuous need for high quality training datasets with accurate labels.

Enhanced CASA holds significant potential for improving fertility analysis, largely because of its automation and standardization, which remains problematic for analyses that rely on the subjective performance of laboratory technicians [34,35]. Despite its potential, CASA has historically underperformed in its predictive value for assisted reproductive therapies and diagnostic tests, often necessitating additional functional assays [36]. Currently, CASA is most useful for relatively extreme phenotypes such as low sperm counts (oligozoospermia), completely immotile sperm (asthenozoospermia), or obviously malformed sperm (teratozoospermia) but lacks discriminatory ability when sperm function or morphology is subtle or context dependent [37]. For example, sperm motility patterns in non-viscous fluids differ significantly from more physiologically relevant viscous or visco-elastic fluids [6]. Sperm that appear to move normally in the non-viscous fluids recommended in the WHO standard, may fail to move appropriately in visco-elastic fluids characteristic of the female reproductive tract or the vestments of the egg.

Another important limitation to CASA analysis is that default tracker configurations may significantly alter results, as demonstrated in this report in experiment 1. Sperm are typically classed into motility types (e.g., progressive, hyperactive) for fertility analysis, but apparent motility profile of a sperm sample can change substantially depending on algorithmic parameters such as search radius, maximum frame gap for object loss, or trajectory linking rules [17]. These choices determine whether a moving object is linked into a continuous track or fragmented into multiple short tracks, which in turn alters the derived kinematic parameters and even whether a sperm trajectory is accurately assigned. Tracking program defaults can be a significant strength of CASA, because they enable far more rigorous standardization than lab technicians can provide. However, if applied without quantitative measures that quantify program performance, two laboratories analyzing the same dataset may arrive at different results simply due to differences in tracker configuration [38]. A general lack of transparency regarding algorithmic configuration limits CASA reproducibility and undermines comparisons across studies. Addressing this issue will require not only standardized reporting of tracker settings but also the development of sperm-specific benchmarks that define biologically appropriate parameter ranges, analogous to hyperparameter optimization in broader machine learning practice [39].

Another important source of variability in CASA output is the timescale of analysis. Most CASA implementations rely on relatively short tracks, often spanning less than one to two seconds of observation [5]. This introduces two major problems: 1) sperm motility is summarized via ensemble averaging despite observation over a period of time that represents only a miniscule fraction of the time required to fertilize an egg, and 2) this practice produces noisy estimates of kinematic parameters because the variance is computed for the sperm sample, rather than individual trajectory time-series [2]. Additionally, short timescale observations may be computationally convenient, but are biologically misleading, since sperm motility is highly dynamic and characterized by transitions between progressive movement, periods of quiescence, and hyperactivation that unfold over much longer timescales [6,7]. Capturing the biological variation in these transitions may yield more stable kinematic measures, but doing so also presents practical challenges: sperm frequently cross paths or move out of focus in dense samples leading to limited track continuity. As a result, CASA systems often prioritize feasibility over biological fidelity by diluting sperm samples or by tracking for shorter periods of time - compromises that likely contribute to the limited predictive power for more subtle fertility outcomes. These timescale dependent tracking limitations are closely intertwined with the methods used for cell detection, since reliable long-term tracking ultimately depends on the accuracy and stability of the underlying detection step. Improved methods are an absolute necessity for CASA to reach it full predictive potential.

The accuracy of CASA output ultimately depends on the method of detection, since tracking quality cannot exceed the fidelity of the initial object identification. Most current systems and training datasets, such as VISEM-tracking, rely on bounding-box based approaches [18]. While computationally efficient, these methods introduce substantial error: bounding boxes are sensitive to orientation changes and background noise, and inaccurately placed bounding box centers creates artifactual noise in kinematic parameter calculations as demonstrated in experiment 3 of this report.

These limitations are particularly problematic for the phase-contrast imaging typical of CASA due to 'halo' effects which complicate bounding box placement. In dense samples, overlapping trajectories further exacerbate misidentification, leading to spurious tracks and inflated motility metrics. Overcoming these challenges will require more sophisticated detection methods, potentially combining deep learning–based segmentation with physics-informed constraints on trajectory smoothness and flagellar-beat periodicity. This approach has been applied previously, but remains relatively low throughput and lacks an open-source codebase limiting its application [40]. Framing detection, tracking, and timescale of analysis as interdependent problems will provide a clear path toward next-generation CASA systems with high predictive value for use in clinical andrology and elsewhere.

Several limitations of this study should be acknowledged. First, our labeled dataset is relatively small and is simply intended as a proof-of-principle to highlight the importance of long timescale tracking. Additionally, the samples do not represent complex changes that sperm undergo during in vitro capacitation, nor was fertility of the samples directly assessed. The ground truth trajectories were derived by manual correction of Trackpy outputs, an approach that introduces potential bias from the initial algorithm and emphasizes head centroids rather than full cell morphology. Imaging constraints, including relatively low frame rates (necessitated by tracking duration), low magnification, and phase-contrast artifacts also contribute noise that propagates into both kinematic parameters and tracking metrics. In addition, while we adapted well-established multi-object tracking metrics and distributional comparisons, these measures may not fully capture sperm-specific features such as flagellar beat periodicity or long-term motility pattern phase transitions linked to capacitation. Despite these limitations, the framework presented here provides a transparent and extensible foundation highlighting the importance of releasing open data and code in standardized formats. Our aim is to enable iterative improvements by the community and to encourage future studies that address these challenges through larger, more diverse datasets, multimodal imaging, and metrics tailored specifically to the unique challenges of sperm video-microscopy.

## Conclusion

Taken together, these results point toward the need for a new framework in CASA that treats detection, tracking configuration, and timescale of analysis in an interconnected way. The limitations highlighted in this report such as variability in tracker configuration, short observation windows, and imprecise detection methods are fundamental barriers to reproducibility and predictive value of CASA. Overcoming them will require three key advances: 1) the establishment of standardized benchmarks and biologically relevant metrics to compare algorithms, 2) the creation of high-quality accurately labeled training datasets that reflect the unique challenges of sperm morphology and imaging artifacts, and 3) the development of integrated pipelines that combine machine learning approaches with physics-informed constraints on sperm motion. Such a framework would transform CASA from a descriptive tool that reports kinematic averages into a predictive system capable of informing clinical decision-making and improving outcomes in assisted reproduction. By adopting open-source, transparent codebases and fostering reproducible methods, CASA could finally deliver on its promise to standardize semen analysis in the same manner that computer vision has revolutionized diagnostics in radiology and other fields.

With this goal in mind, we make our code publicly available for others to use and run their methods on. The source code can be found at github.com/CAS-ReproLab/Sperm_Object_Tracking. For a given video, trajectory data can be stored in a single comma-separated-values file (.csv). This .csv file contains x,y coordinates, a sperm label, and frame in each row. This provides an efficient, human-readable, and unambiguous way of storing predictions and/or ground truth data. Once the prediction data is provided in a standard format, the metrics can be computed to evaluate its performance as previously described.

In future work, we aim to expand our comparisons to additional datasets such as MIaMIA-SVDS [41]. Each dataset stores the trajectory information in a different way, so tools will need to be developed to convert between representations.

Additionally, though the segmentation is not ultimately needed for trajectory tracking, obtaining full sperm outlines can be helpful for biological outputs such as flagellar waveform analysis. Additional baselines and metrics can be established to properly account for these use cases.

## Supporting information

**S1 Appendix. Mathematical formulation of common cell tracking metrics.**
(PDF)

## Author contributions

**Conceptualization:** David Hart, Kylie Cashwell, Anita Bhandari, Cameron Schmidt.

**Data curation:** Kylie Cashwell, Jayath Premasinghe, Cameron Schmidt.

**Formal analysis:** David Hart, Kylie Cashwell, Jayath Premasinghe, Cameron Schmidt.

**Funding acquisition:** Cameron Schmidt.

**Investigation:** Cameron Schmidt.

**Methodology:** David Hart, Anita Bhandari, Jayath Premasinghe.

**Project administration:** Cameron Schmidt.

**Resources:** Cameron Schmidt.

**Software:** David Hart, Kylie Cashwell, Anita Bhandari, Jayath Premasinghe.

**Supervision:** David Hart, Cameron Schmidt.

**Validation:** David Hart, Kylie Cashwell, Anita Bhandari, Cameron Schmidt.

**Visualization:** David Hart, Kylie Cashwell, Anita Bhandari, Cameron Schmidt.

**Writing – original draft:** David Hart, Cameron Schmidt.

**Writing – review & editing:** David Hart, Kylie Cashwell, Anita Bhandari, Jayath Premasinghe, Cameron Schmidt.

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
