## [Decision Letter · Decision Letter 0]

13 Nov 2025

PCOMPBIOL-D-25-01941

A framework for evaluating predicted sperm trajectories in crowded microscopy videos

PLOS Computational Biology

Dear Dr. Hart,

Thank you for submitting your manuscript to PLOS Computational Biology. After careful consideration, we feel that it has merit but does not fully meet PLOS Computational Biology's publication criteria as it currently stands. Therefore, we invite you to submit a revised version of the manuscript that addresses the points raised during the review process.

We look forward to receiving your revised manuscript.

Kind regards,

Jing Chen

Academic Editor

PLOS Computational Biology

Dimitrios Vavylonis

Section Editor

PLOS Computational Biology

**Additional Editor Comments:**

The reviewers complimented the value and novelty of the method. However, there are mixed opinions about the clarity of the manuscript, as well as ambiguity about the validation sample size. Please address these concerns in the revision.

**Journal Requirements:**

At this stage, the following Authors/Authors require contributions: David Hart. Please ensure that the full contributions of each author are acknowledged in the "Add/Edit/Remove Authors" section of our submission form.

3) We noticed that you used the phrase 'data not shown' in the manuscript. We do not allow these references, as the PLOS data access policy requires that all data be either published with the manuscript or made available in a publicly accessible database. Please amend the supplementary material to include the referenced data or remove the references.

5) We notice that your supplementary information is included in the manuscript file. Please remove them and upload them with the file type 'Supporting Information'. Please ensure that each Supporting Information file has a legend listed in the manuscript after the references list.

State the initials, alongside each funding source, of each author to receive each grant. For example: "This work was supported by the National Institutes of Health (####### to AM; ###### to CJ) and the National Science Foundation (###### to AM).".

**Reviewers' comments:**

Reviewer's Responses to Questions

**Comments to the Authors:**

Reviewer #1: Dear authors,

The topic of your research is interesting and original. There are, however, some comments mainly about the organization of the manuscript. The experimental design of the study is not described in the materials and methods part, but it seems that this information is included in the results part. For instance:

- L100: Tracking Quality’s Effect on Standard Sperm Analysis Outputs

This section could be included in the introduction part or could be discussed/compared/contrast with the results in the discussion.

- Line 129: “We then provide three experiments that illustrate poor tracking assumptions and the effects on the calculated kinematic parameters.”

This sentence could be the final in the introduction. The three experiments are not described in the materials.

- The following sections: Baseline Tracker and Dataset (L132), Experiment 1: A Poorly Configured Tracker (L152), Experiment 2: Short vs. Long Timescale Trajectories (L177), Experiment 3: Bounding Boxes vs. Centroids (L204), Evaluating Sperm Detection and Tracking (L235), Unique Considerations for Evaluating Sperm Detection (L269), Unique Consideration for Evaluating Sperm Tracking (L301), Evaluation on the Dataset (L337-44)

In the above mentioned paragraphs information about the experimental design are included in.

- The sentence “We also output Earth Mover’s Distance (EMD) scores for each kinematic parameter for the suboptimal and optimal approach when compared to the ground truth (L368-69)

This approach could be included in the materials as part of the corresponding step.

Please, re-organize the results and materials and methods part.

Reviewer #2: The manuscript presents a computational framework for evaluating sperm tracking algorithms in crowded microscopic fields. The topic is relevant and the hypothesis sound strong. The manuscript is well structured and written clearly. I enjoyed reading it, up the number of videos thats being assessed. However, there are some issues needs to be clarify.

1. The dataset used for validation is too small. Even though they used several tracking system. You could give the number of spermatozoa that are being evaluated instead of number of videos. This details should be given in the abstract.

2. The proposed metrics are adapted from existing tracking systems, so their novelty must be clearly defined in the methods.

3. The biological implications of improved tracking must be discussed more cautiously, as they didnt induced capacitation nor the fertility.

Overall, the novelty of the paper sound. However, the limitations of the study must be clearly given in the abstract and the discussion. At least why authors chose 3 videos for validating the existing tracking system such as due the number of sperm trajectories, time manner etc. For instance, whether this was due to number of trajectories, frame duration, ot computational limitations. Although, analysing longer videos provides richer motion data than traditional casa systems that are typically tracks sperms for 1-2 sec). This approach must be clearly explained if the dataset contains the number of spermatozoa that are being evaluated (300 spermatozoon), instead of number of videos. Therefore, the authors may indicate the total number of trajectories analyzed and how many overlapping trajectories are detected, rather than simplifiying the number of videos. Otherwise, this dataset may create ambiguity for the readers. This might change the overall assumptions more trustworthy.

**Have the authors made all data and (if applicable) computational code underlying the findings in their manuscript fully available?**

Reviewer #1: Yes

Reviewer #2: Yes

PLOS authors have the option to publish the peer review history of their article (what does this mean?). If published, this will include your full peer review and any attached files.

Reviewer #1: No

Reviewer #2: No

**Figure resubmission:**
---

## [Decision Letter · Decision Letter 1]

28 Jan 2026

Dear Dr. Hart,

We are pleased to inform you that your manuscript 'A framework for evaluating predicted sperm trajectories in crowded microscopy videos' has been provisionally accepted for publication in PLOS Computational Biology.

Best regards,

Jing Chen

Academic Editor

PLOS Computational Biology

Dimitrios Vavylonis

Section Editor

PLOS Computational Biology

Reviewer's Responses to Questions

**Comments to the Authors:**

Reviewer #1: Dear authors,

The manuscript is clear and well documented, with all the necessary information and explanations included.

Thank you for your cooperation.

Reviewer #2: The authors have carefully addressed the comments and the revised version is suitable for publication.

**Have the authors made all data and (if applicable) computational code underlying the findings in their manuscript fully available?**

Reviewer #1: Yes

Reviewer #2: Yes

PLOS authors have the option to publish the peer review history of their article (what does this mean?). If published, this will include your full peer review and any attached files.

Reviewer #1: No

Reviewer #2: No

---

## [Editor Report · Acceptance letter]

PCOMPBIOL-D-25-01941R1

A framework for evaluating predicted sperm trajectories in crowded microscopy videos

Dear Dr Hart,

I am pleased to inform you that your manuscript has been formally accepted for publication in PLOS Computational Biology. Your manuscript is now with our production department and you will be notified of the publication date in due course.

With kind regards,

Lilla Horvath
